# Current Understanding of Potential Linkages between Biocide Tolerance and Antibiotic Cross-Resistance

**DOI:** 10.3390/microorganisms11082000

**Published:** 2023-08-03

**Authors:** Kent Coombs, Cristina Rodriguez-Quijada, Jason O. Clevenger, Alexis F. Sauer-Budge

**Affiliations:** Exponent, Inc., Biomedical Engineering & Sciences, Natick, MA 01760, USA; kcoombs@exponent.com (K.C.);

**Keywords:** antimicrobial, biocide tolerance, antibiotic cross-resistance

## Abstract

Antimicrobials (e.g., antibiotics and biocides) are invaluable chemicals used to control microbes in numerous contexts. Because of the simultaneous use of antibiotics and biocides, questions have arisen as to whether environments commonly treated with biocides (e.g., hospitals, food processing, wastewater, agriculture, etc.) could act as a reservoir for the development of antibiotic cross-resistance. Theoretically, cross-resistance could occur if the mechanism of bacterial tolerance to biocides also resulted in antibiotic resistance. On the other hand, biocides would likely present a higher evolutionary barrier to the development of resistance given the different modes of action between biocides and antibiotics and the broad-based physicochemical effects associated with most biocides. Published studies have shown that the induction of biocide tolerance in a laboratory can result in cross-resistance to some antibiotics, most commonly hypothesized to be due to efflux pump upregulation. However, testing of environmental isolates for biocide tolerance and antibiotic cross-resistance has yielded conflicting results, potentially due to the lack of standardized testing. In this review, we aim to describe the state of the science on the potential linkage between biocide tolerance and antibiotic cross-resistance. Questions still remain about whether the directed evolution of biocide tolerance and the associated antibiotic cross-resistance in a laboratory are or are not representative of real-world settings. Thus, research should continue to generate informative data to guide policies and preserve these tools’ utility and availability.

## 1. Introduction

The term “antimicrobial” is used to describe a broad set of chemical agents that are used to help control the spread of microbes in a variety of applications. Antimicrobials can be split into two main categories: (1) antimicrobial biocides, which are used in a variety of contexts, including but not limited to antiseptics, surface disinfectants, material preservatives, and/or water-recycling treatments, and (2) antimicrobial drugs (e.g., antibiotics), which are utilized to treat human or animal infections [1,2]. Biocides are unequivocally important to modern human society, with widespread use in household products, food preservatives, agriculture, and clinical settings, where they play a key role in controlling pathogens [3,4]. Biocides have a long history, starting with early examples such as using copper vessels for potable water storage, vinegar and iodine for wound treatment, and phenol (carbolic acid) in antiseptic surgeries [4]. Other biocides were introduced in the first half of the 20th century, including chlorine-releasing agents and some quaternary ammonium compounds. Antibiotics are also indispensable to our society and have been credited for the extension of the human lifespan as a result of their use across the world [5]. In addition to treating infections in humans, antibiotics are used prophylactically and to treat infections in pets and livestock [6]. Antibiotics isolated from various microorganisms were introduced in the 1930s and 1940s to treat human infections, including sulphonamides, penicillin, and streptomycin [5].

Because of the importance of antibiotics in modern medicine, the emergence and proliferation of antibiotic resistance have become an issue of increasing concern in our society [6,7]. Moreover, the emergence of multidrug resistant (MDR) bacteria that can evade the effect of at least one antibiotic in three or more drug classes has led to increased efforts to understand and control the proliferation and emergence of MDR strains [8]. Bacteria have evolved a variety of strategies that enable resistance to these drugs, including new cellular processes to evade the antibiotic effect, enzymes to modify the antibiotic, restrictions to access antibiotic targets, and pumps to eject antibiotics. Although these mechanisms are commonly referred to as “antimicrobial resistance”, the discussion primarily focuses on antimicrobial drugs, specifically antibiotics. Thus, for the purpose of this review, we will focus on antibiotic resistance and refer the reader to other reviews for information on antifungal [9] or antiviral drugs [10,11].

Similar to the concerns regarding antibiotic resistance, concerns have arisen with respect to the potential of bacteria to evade the effects of biocides. Although less well studied, tolerance to a variety of biocides has been reported (e.g., [4,12,13,14,15,16,17,18,19,20,21,22,23,24,25]). The reported mechanisms that bacteria use to reduce the impact of biocides include enzymes to modify the biocide, changes in the permeability of the membrane, and efflux pumps to reduce the intracellular concentration of the biocide. It is important to note that unlike antibiotic resistance, where standard methods and definitions exist to measure and define efficacy with respect to clinical therapeutic usage, currently, there are no standard methods or definitions to qualify or quantify biocide efficacy. Instead, a diversity of terms are used with biocides, such as “resistance”, “tolerance”, “decreased susceptibility”, and “reduced susceptibility” [12]. Because many of the reported instances of reduced effectiveness of biocides are at concentrations significantly below the specified in-use concentrations, we will use the term biocide tolerance so as not to imply that these changes equate to bacterial survival at in-use concentrations.

In addition to the first-order concern regarding a potential increase in biocide tolerance, a second-order concern has arisen. Due to the use of biocides alongside antibiotics, e.g., clinical settings and animal husbandry, it has been hypothesized that biocide usage may provide selective pressure that results in antibiotic cross-resistance. For this hypothesis to be accurate, the bacterial mechanisms to evade biocides must be the same as those used to evade antibiotics; thus, in this review, we summarize the chemistries, modes of action, and known resistance/tolerance mechanisms for major classes of antibiotics and biocides to highlight areas of similarity and differences. Then, we review the literature, investigating potential links between biocide usage and antibiotic cross-resistance and conclude with a discussion of the current body of evidence. While some reported mechanisms relate to the intrinsic structural properties of bacteria, such as permeability differences between Gram-positive and Gram-negative bacteria or due to biofilm formation, we have not focused on these mechanisms in our analysis and refer the reader to other reviews that are focused on those topics [26,27,28].

## 2. Antibiotics—Major Drug Classes, Chemistries, Modes of Action, and Resistance Mechanisms

Antibiotics are a sub-type of antimicrobial drugs, which are used therapeutically to control infections in humans and animals. As such, antibiotics must act on specific bacterial targets that are sufficiently different from those found in eukaryotic cells to avoid toxicity to the patient. Currently marketed antibiotics target cell wall synthesis, protein synthesis, nucleic acids (DNA/RNA), metabolic pathways, and the cell membrane due to the specificity of the antibiotic modes of action (Table 1) [29]. Specific chemistries of the antibiotics enable their targeted modes of action, for example, binding to an active site of a key enzymatic process. Likewise, resistance mechanisms are also commonly quite specific to that antibiotic and/or class of antibiotics. Generally, antibiotic resistance mechanisms fall into several main categories (Figure 1) [30,31]:Alteration of the target thereby preventing the drug from binding;Enzymatic modification of the drug to degrade or modify it;Decrease in the accumulation of the antibiotic by the alteration of porins (reducing access) or by the overexpression of efflux transporters (increasing removal);Overproduction of the target to overwhelm the drug.

The mode of action and examples of antibiotic resistance mechanisms are briefly described in this section.

### 2.1. Antibiotics That Target Cell Wall Biosynthesis

Two major classes of antibiotics target cell wall biosynthesis, those based on the β-lactam ring structure and glycopeptides. All β-lactams share a similar mode of action where they primarily act as transpeptidase inhibitors and thereby impair cell wall biosynthesis [29,33]. Resistance to β-lactams occurs by three main mechanisms: (1) modification of the transpeptidase target; (2) production of β-lactamases and carbapenemases (hydrolyzing enzymes); or (3) decrease in the accumulation of the antibiotic by the alteration of porins (reducing access) or by the overexpression of efflux transporters [29,31,33,39,40].

Glycopeptides, such as vancomycin, interfere with cell wall biosynthesis by binding to precursors within the cell wall and thereby preventing the addition of new units to the peptidoglycan. Resistance to glycopeptides results from a modification of the precursor that reduces the affinity of the antibiotic to its target [29,31].

### 2.2. Antibiotics That Target Protein Synthesis

A variety of antibiotic classes target the inhibition of protein synthesis, including aminoglycosides, tetracyclines, macrolides, lincosamides, chloramphenicol, and oxazolidinones [29,35]. The antibiotics that interfere with protein synthesis do so by binding to either the 30S or 50S ribosomal subunit or by interfering with the initiation of the ternary complex of the 30S and 50S ribosomal subunits. Resistance to antibiotics that target protein synthesis occurs through the production of antibiotic-modifying enzymes, changes to membrane permeabilization (increased expression of efflux pumps or decreased expression of porins), and alterations to the antibiotic binding site [40,41,42].

### 2.3. Antibiotics That Affect Nucleic Acids

Examples of antibiotics that affect nucleic acids include fluoroquinolones, ansamycins, and lipiarmycins. Fluoroquinolones inhibit the activity of topoisomerases, including enzymes that supercoil DNA (DNA gyrases) and those that relax supercoiled DNA (topoisomerase IV) [29]. Resistance to fluoroquinolones is known to have chromosomally mediated mechanisms, such as topoisomerase mutation, loss or expression of porins (e.g., *OmpA*), or increased expression of efflux pumps [40,43,44]. Plasmid-mediated resistance has also been described, including the production of *Qnr* proteins (DNA gyrase protection), AAC(6′)-Ib-cr (modifies ciprofloxacin), and the plasmid-encoded efflux pumps (e.g., *QepA* and *OqxAB*) [44,45].

Ansamycins (e.g., rifampin) and lipiarmycins (e.g., fidaxomicin) act on the RNA polymerase and thereby inhibit DNA transcription. The primary mechanism for rifampin and fidaxomicin resistance is caused by mutations in the gene that encode for the β-subunit of the bacterial RNA polymerase (*rpoB*) [46,47,48]. Resistance to rifampin has also been shown to be conferred through reduction in the permeability of the cell wall and through the expression of efflux pumps [46,49].

### 2.4. Antimetabolite Antibiotics

Sulfonamides and diaminopyrimidines are antimetabolite antibiotics that inhibit the folate pathway in bacteria [50]. Sulfonamides inhibit dihydropteroate synthase (DHPS) through a higher affinity for the enzyme as compared to its natural substrate, p-aminobenzoic acid. Diaminopyrimidines, such as trimethoprim, bind to dihydrofolate reductase (DHFR). Resistance to antifolates is known to occur through the hyperproduction of p-aminobenzoic acid or by mutations that alter the enzyme affinity for the antibiotic [29,31].

### 2.5. Antibiotics That Target the Membrane

Antibiotics that target the membrane include the lipopeptide and cyclic polypeptide classes. Lipopeptides, such as Daptomycin, form micelles (oligomeric assemblies) that interact with the membrane to cause leakage of cytosolic contents, while cyclic polypeptides, such as polymyxins and colistins, act as detergents and alter the permeability of the membrane [29,37,38]. Identified resistance mechanisms to these classes of antibiotics primarily relate to modifications to the composition of the cell membrane through lipopolysaccharide remodeling and the overexpression of certain efflux pumps (e.g., *AcrAB–TolC*) [38].

### 2.6. Antibiotics Summary

Bacteria use a variety of mechanisms to evade antibiotics. Due to the specificity of the modes of action, many of the resistance mechanisms are also very specific to the antibiotic, such as through modification of the antibiotic or mutations in the binding pocket. Some of the resistance mechanisms are more generalized, such as the expression of efflux pumps, which may eject other substances in addition to the antibiotics. To compare antibiotics to non-drug antimicrobials, we next summarize the major classes of biocides through a description of their chemistry, modes of action, and resistance mechanisms.

## 3. Biocides—Major Classes, Chemistries, Modes of Action, and Resistance Mechanisms

Although their modes of action are not fully understood, biocides generally act on multiple targets within the bacteria in a non-selective manner, such as through ionic interactions, the disruption of hydrogen bonding, and chemical reactions (such as oxidants and electrophiles) (Table 2) [26,51,52,53,54,55]. The generality of biocidal action is due to a fundamental difference in chemistry between antibiotics and biocides. The chemical modalities of biocides are not specific to a particular biochemical pathway, but instead can act on multiple structural and functional components of the bacteria, thereby disrupting cell walls, cell membranes, proteins, and nucleic acids. These mechanisms undermine the fundamental drivers of the tertiary and quaternary structures of biological macromolecules, which explains their widespread disruption of bacterial pathways. Therefore, the emergence of biocide resistance is unlikely to be caused by specific alterations of the target site or by overproduction of the target site to overwhelm the effect of the biocide, as is seen in antibiotic resistance [12,56]. One notable counter-example to the non-selective modes of action is triclosan, which has been shown at low concentrations to be a site-specific inhibitor of enoyl-acyl carrier protein reductase, and targeted resistance has been reported [57,58]. More commonly, generalized mechanisms that decrease the accumulation of biocides within the bacteria by altering the permeability of the membrane or by the overexpression of particular efflux transporters have been reported [40,58,59]. Enzymatic transformation of some biocides has also been reported, e.g., heavy metals and formaldehyde [60].

### 3.1. Biocides That Inactivate through Ionic Interactions

#### 3.1.1. Quaternary Ammonium Compounds (QACs)

QACs are cationic molecules whose positively charged molecules bind strongly to cell walls and membranes, and their mode of action stems from their ability to interact electrostatically with phospholipids [26,51,52,53,61,62,68]. Efflux pumps have been identified as a potential mechanism of QAC tolerance [26,51,52,53,61,62,68].

#### 3.1.2. Bisbiguanides

Bisbiguanides are also categorized as cationic antimicrobials [68]. They work by crossing/damaging the cell wall/membrane, and subsequently causing cytoplasmic coagulation and enzyme disruption, as well as DNA disruption through electrostatic interactions with phospholipids [15,26,52,53,62]. It has been hypothesized that acquired tolerance to chlorhexidine (one type of bisbiguanides) might be linked to the overexpression of efflux pumps or the acquisition of plasmid-encoded efflux pumps [53,64].

### 3.2. Biocides That Inactivate through the Disruption of Hydrogen Bonds

#### 3.2.1. Phenolics

Phenolics’ general mode of action is not fully understood, but it has been proposed that they induce changes in membrane permeability and intracellular functions through hydrogen bonding. One particular phenolic, triclosan (TRI), has been shown at low concentrations to act as a site-specific inhibitor of enoyl-acyl carrier protein reductase [17,57,69]. The upregulation of an enoyl reductase (FabI) and efflux pumps are thought to be the main mechanisms of triclosan tolerance [14,69,70,71,72,73].

#### 3.2.2. Alcohols

The general mechanism for alcohols includes the coagulation/degradation of proteins and lipids with water-dependent activity to permeate cell membranes. The mode of action of alcohols is understood to be the dissolution of phospholipids and denaturation of proteins through the disruption of hydrogen bonding [26,51,62]. We were unable to identify any bacterial tolerance mechanisms to alcohols in the literature.

### 3.3. Biocides That Inactivate through Chemical Reactions

#### 3.3.1. Metals

Biocides based on heavy metals (e.g., copper and silver salts) are understood to interact with the thiol groups on proteins, such as cytoplasmic and membrane-bound enzymes, and thereby causing metabolic inhibition [26,53,54,62]. The overexpression of efflux pump proteins and a reduced expression of porins have been described as possible mechanisms of metal tolerance [74,75], as well as an enzymatic reduction of the cation to the metal [60,70]. Generally, authors have reached the agreement that the exact mechanisms still remain unclear and are also organism specific [76,77,78,79]. Therefore, further investigation is needed.

#### 3.3.2. Chlorine-Releasing Agents

Released chlorine causes cell membrane damage by protein and lipid oxidation and can also inhibit and degrade DNA and RNA [23,54]. Chlorine-releasing compounds are understood to halogenate amino groups in proteins as well as oxidize thiol groups, resulting in metabolic inhibition and lysis [51,59]. Reduced susceptibility to chlorine-releasing compounds has been shown via intrinsic mechanisms of biofilm formation or from certain spore coats, e.g., *B. subtilis* spores with α/β-type small acid-soluble spore proteins [26,53,60]. The upregulation of the *acrF* gene, which encodes the ACrEF efflux pump, was also observed by Curiao et al. The authors concluded that the mechanism of cross-resistance is likely multi-factorial as a result of the complex variety of antimicrobial mechanisms that affect multiple basic networks of bacterial physiology [13].

#### 3.3.3. Fixatives (Aldehydes)

Formaldehyde is a very effective biocide, damaging cells by interacting with the cell membrane and cytoplasmic proteins as well as intramolecular and intermolecular cross-linking of molecules, but its use has been limited due to its high toxicity [80]. The biocidal activity of formaldehyde and glutaraldehyde results from the alkylation of biomolecules with amino, imino, amide, carboxyl, and thiol groups on proteins and nucleic acids [51,53,59]. Expression changes in dehydrogenases have been shown in tolerant phenotypes, including adhC in *E. coli* [64,81]. Reduced susceptibility can also be achieved by the enzymatic transformation of formaldehyde into non-toxic products [12,60,70]. Additionally, changes in porin expression have been associated with increased aldehyde tolerance of *Mycobacterium* [82]. Formaldehyde-releasing agents are still commonly used as preservatives. Bronopol, which is thought to release formaldehyde, is discussed further below [83].

#### 3.3.4. Peroxygens

Hydrogen peroxide acts by producing hydroxyl free radicals that degrade various cellular components, e.g., enzyme and protein thiols, and peracetic acid is suspected to have a similar mode of action [26,51,53,54,62,64]. Enzymatic degradation of peroxygen compounds has been proposed as the primary tolerance mechanism [64,70]. It has also been shown that small acid-soluble spore proteins in *B. subtilis* spores contributes to the spore tolerance to peroxide [53].

#### 3.3.5. Iodine

Iodine acts by quickly penetrating the cell wall and oxidizing key cellular components, including thiol groups on proteins, as well as oxidizing nucleotides and fatty acids [53,54,56,64,65]. Povidone iodine is known to have variable activity against some Actinobacteria (e.g., *Corynebacterium* spp. and *Mycobacterium* spp.) due to the high mycolic acid content of their cell walls, which makes it difficult for free iodine to penetrate [84]. To our knowledge, no transferrable tolerance mechanisms have been described in the literature, although recalls have been reported with potential biological contamination of some povidone iodine products [64,85].

#### 3.3.6. Bronopol

Bronopol is known to react with thiol groups on cytoplasmic and membrane-bound enzymes, e.g., dehydrogenases, which results in metabolic inhibition [62]. It is also associated with low levels of formaldehyde release, although it is not formally regarded as a formaldehyde releaser by some authorities [83]. Under aerobic conditions, bronopol has been shown to catalytically oxidize thiol-containing proteins (e.g., cysteine), resulting in superoxide and peroxide by-products, which in turn are responsible for its bactericidal activity [66]. Limited information is available on bronopol resistance, although it has been hypothesized that quorum sensing might have a role in tolerant phenotype establishment and biofilm formation [59,86].

#### 3.3.7. Ethylene Oxide

Ethylene oxide (EtO) is an alkylating agent that is known to attack amino and thiol groups in proteins, as well as DNA and RNA [26,67]. To our knowledge, no tolerance mechanisms have been identified for ethylene oxide.

#### 3.3.8. Isothiazolinone

The antibacterial properties of isothiazolinones are understood to be due to their ability to act as an electrophilic agent that reacts with critical enzymes, with thiols on proteins, and with the production of free radicals [26,55]. To our knowledge, no information on isothiazolinone tolerance has been described in the literature [64].

### 3.4. Biocides Summary

In contrast to antibiotics, less is known about the bacterial mechanisms that confer biocide tolerance. For some biocides, no tolerance mechanisms have been described in the literature. Due to the non-selective and multifactorial nature of the biocide modes of actions, the majority of the tolerance mechanisms described are not unique to a particular biocide, such as efflux pumps or changes in porin expression. Chemistry-specific tolerance mechanisms have been described in limited cases, such as enzyme degradation, as well as for triclosan with changes in the *fabI* gene. In the next section, we explore the potential connection between biocide tolerance and antibiotic cross-resistance.

## 4. Summary of Studies Investigating the Potential for Antibiotic Cross-Resistance

Due to some similarities between the antibiotic resistance and biocide tolerance mechanisms, as well as the use of both in certain contexts, such as healthcare, animal husbandry, and food production, concerns have been raised that the use of biocides may result in antibiotic resistance and subsequent treatment failure. For this hypothesis to be true, the mechanisms evoked by bacteria to evade the impact of biocides must be promiscuous, such that antibiotics with very different chemistries and targets are nullified. As discussed in the previous sections, a variety of mechanisms are used to increase biocide tolerance that have some similarities with those for antibiotic resistance. While bacteria have been shown to enzymatically degrade certain biocides (e.g., formaldehyde and peroxides) and a number of antibiotics (e.g., β-lactamases), the mechanisms are chemistry-specific and therefore are not expected to infer cross-resistance [12,29,31,33,39,40,60,64,70,81]. The inherent difference in the specificity of the targets of biocides and antibiotics also limits the risk of cross-resistance. Overproduction of the target can reduce the susceptibility of bacteria to antibiotics, but this mechanism cannot be effective for biocides due to their non-selectivity. Likewise, while bacteria can evade the effectiveness of antibiotics through relatively minor changes to the target, such as a mutation in the binding site, bacteria cannot use this type of minor change to evade biocides. One exception is that triclosan has been shown at low concentrations (0.02–0.5 mg/L) to have a specific target, which is also a target for the antimycobacterial drug isoniazid [17]. At higher concentrations (5–35 mg/L), triclosan has more broad impacts on the cells, and therefore, a mutation of the enoyl-ACP reductase alone is not expected to inhibit triclosan at in-use concentrations [87]. However, changing the accumulation of biocides in the bacteria through a reduction in access (porins) or increased efflux (efflux pumps) may theoretically be able to infer cross-resistance.

In this section, we review the literature, exploring potential linkages between biocide use and a causal relationship to antibiotic cross-resistance. The majority of the literature is focused on QACs, bisbiguanides (chlorhexidine), and phenolics, while comparatively less information was found on metals, chlorine-releasing agents, fixatives (glutaraldehyde and formaldehyde), peroxygens (hydrogen peroxide and peracetic acid), alcohols, and iodine. We have summarized the major findings for each of these biocide categories in the subsections below. Since no relevant information was found for bronopol, DDBSA, ethylene oxide, or isothiazolinones, these biocides are not discussed further.

The literature was found to be divided into two major types of studies. Much of the literature focused on studies in which bacteria strains were subjected to increasing sub-inhibitory concentrations of biocide over multiple generations with the goal of eliciting an adaptive response leading to increased biocide tolerance. At the end of this process, the bacteria with increased biocide tolerance were then assessed to see if a corresponding antibiotic resistance could be measured. Another less explored area of research has been the assessment of bacterial isolates from environmental samples such as hospital surfaces, food processing areas, wastewater, mines, agriculture, and lakes. The goal of this type of study is to acquire more “real-world” evidence as to whether bacteria are tolerant to a biocide and then assess if they are cross-resistant to one or more antibiotics.

One challenge in reviewing the literature is that completely different methods are used to determine the in-use concentrations of antibiotics and of biocides [88]. The in-use concentration for antibiotics are related to the therapeutic dose used to treat a bacterial infection in vivo. Antibiotics are designed for use in live tissues to enable the immune system of the host to gain control over the infection. The bacterial susceptibility or resistance to a particular antibiotic is generally assessed by inoculating a bacterial isolate with different concentrations of the antibiotic to determine the minimum inhibitory concentration (MIC) through standardized protocols. The experimentally measured MIC is then compared to standardized breakpoints established by standards organizations like the European Committee on Antimicrobial Susceptibility Testing (EUCAST) and the Clinical and Laboratory Standards Institute (CLSI) to determine if the isolate is clinically susceptible, clinically intermediate, or clinically resistant [89,90]. In contrast, while MICs can be determined for biocides, they are not used as the basis for in-use concentrations. Unlike antibiotics, the purpose of biocides is to kill bacteria swiftly, and relying on MIC measurements can be misleading. Biocide effectiveness is assessed either by time-kill procedures or determination of the concentration that produces a certain log reduction [91,92]. Where possible in our analysis, we considered increased MIC values measured for particular biocides related to recommended in-use concentrations.

### 4.1. Quaternary Ammonium Compounds (QACs)

Several studies focused on inducing QAC tolerance in the laboratory by growing strains at low/subinhibitory concentrations of different QACs spiked into growth media. Although increased QAC tolerance could be induced, in many cases the QAC concentrations remained below the recommended in-use concentration suggested by manufacturers. A few examples were identified in which bacteria were subjected to increasing concentrations of QACs (e.g., benzalkonium chloride—BAC) and eventually developed tolerance that exceeded the recommended BAC in-use concentrations for some applications, e.g., in an alcohol-free hand sanitizer (1000 mg/L or 0.1%), but not others (e.g., diluted shampoo (5000 mg/L or 0.5%)) [93,94]. The bacteria in this study included *Salmonella enterica* serovar Typhimurium (3000 mg/L), *Pseudomonas aeruginosa* (2500 mg/L), *Enterobacter* spp. (1500 mg/L), *Escherichia coli*, and *Staphylococcus saprophyticus* (1000 mg/L). Multiple studies using adaptive evolution techniques with subinhibitory concentrations to develop increased tolerance to QACs (e.g., 2-fold to over 100-fold higher MICs) and also reported cross-resistance or elevated antibiotic MICs to some antibiotics [74,95,96,97,98,99,100,101,102,103,104,105,106,107,108,109]. However, not all of the studies measured antibiotic resistance before adaptive evolution, and others showed increased susceptibility to certain antibiotics after adaptation. Furthermore, the induced cross-resistance to antibiotics was not necessarily stable and could return to wild-type values after continued passages in the presence of the QAC [101].

In studies that identified the development of antibiotic cross-resistance in bacteria tolerant to QACs, efflux pumps were suggested as a possible mechanism of regulation [104,110]. Various efflux genes have been shown to be upregulated after exposure to BAC and other biocides under laboratory conditions; however, there is conflicting evidence as to whether efflux pumps are the main driver of antibiotic cross-resistance [104,105].

Several articles described bacteria that were isolated directly from environments that commonly use QAC-based disinfection. These isolates were first tested for tolerance to specific QACs, and then the identified strains were challenged with antibiotics. Little to no correlation with antibiotic resistance was observed with *Listeria monocytogenes*, *E. coli*, and *Staphylococcus aureus* isolated from food processing plants, fish farms, poultry feces, and clinical settings [111,112,113,114,115,116]. For *P. aeruginosa* isolated from clinical samples, veterinary samples, and wastewater, 23 out of 147 isolates were classified as “resistant” to BAC using an author-derived epidemiological cut-off value of 128 mg/L (0.01% *w*/*v*). The isolates originating from wastewater were more resistant to BAC and demonstrated cross-resistance for fluoroquinolones and multi-dug resistance than those found from other ecological niches [117]. In another study, 87 isolates from seafood were assessed for BAC tolerance, and 5.75% were designated as having high tolerance (≥250 mg/L or 0.02% *w*/*v*) [118]. However, in both of these cases, the concentrations are significantly below in-use concentrations [93,94]. In Condell et al., 189 *Salmonella* strains were tested in seven food industry biocide formulations at in-use concentrations, and only one isolate, *S. enterica*, survived; however, the phenotype was unstable, and the isolate became susceptible with more testing [119]. We did not identify any in-depth comparisons of the fitness of adapted wild-type strains in our review, which may be an important factor in the environment.

Although in vitro studies with QACs have demonstrated the possibility of biocide-induced antibiotic cross-resistance in bacteria, there continues to be a lack of in vivo or in situ studies definitively reporting such a link. Nonetheless, the evidence from in vitro studies demonstrates that antibiotic cross-resistance can be induced under particular laboratory conditions.

### 4.2. Bisbiguanides

Research on bisbiguanides’ potential to induce antibiotic cross-resistance has been focused on chlorhexidine (CH). Despite its common and long history of use, only 14 articles were identified that directly investigated CH’s ability to induce cross-resistance to antibiotics. A few studies demonstrate that various bacteria species were adapted to increase CH tolerance by passaging them in subinhibitory CH concentrations in growth media. The CH concentrations were kept constant or they increased with each passage. Under these conditions, antibiotic cross-resistance was identified in the CH-adapted bacteria [13,104,115,119,120,121]. The investigators studied *Enterococcus faecium*, *Salmonella*, *Klebsiella pneumoniae*, and *S. aureus* and showed new antibiotic cross-resistance to daptomycin, tetracycline, ampicillin, chloramphenicol, cefpodoxime, vancomycin, ciprofloxacin, levofloxacin, and gatifloxacin after CH exposure. Generally, the tolerance to chlorhexidine could be increased 2-fold to 200-fold of the pre-exposure MIC. Similar to the adaptive evolution experiments with QACs, mixed results were reported, in which some strains showed an increase in MICs to antibiotics, while others showed a decrease [97,98].

Two studies evaluated isolates of *Salmonella* and *S. aureus* from different sources, including clinic, food, environment, and water settings. Two out of seven *Salmonella* strains were identified to be tolerant to CH, with a 26- to 51-fold increase in MIC values after several rounds of in vitro selection [119]. However, they were still susceptible to seven food industry biocide formulations at 50% of the manufacturers’ in-use concentration in growth media. A second study looked at 1632 *S. aureus* strains isolated from humans. No bivariate correlations were found between CH exposure and antibiotic cross-resistance [122].

Overall, the body of scientific literature provides evidence that bacteria exposed to subinhibitory CH concentrations in a lab environment can result in antibiotic cross-resistance. However, freshly isolated bacteria from environments with common biocide usage were found to still be susceptible to in-use CH activity without cross-resistance to antibiotics, unless they were adapted in vitro.

### 4.3. Phenolics

Reports investigating the potential development of phenolic tolerance and an associated cross-resistance to antibiotics have focused on triclosan. It has been suggested that because triclosan has a targeted mode of action, it is more likely to induce cross-resistance to antibiotics that share the same targets [28]. However, the importance of laboratory studies where triclosan is used to induce cross-resistance to antibiotics is still debated. Examples of induced cross-resistance to antibiotics, induced sensitivity to antibiotics, and of the lack of correlation between triclosan and antibiotic resistance are all described in the literature. It should also be noted that triclosan is no longer widely used as an active ingredient in biocides, in part due to being banned by the US Food and Drug Administration (FDA) in 2016 for use in antimicrobial soaps used at home by the general population [123].

Several authors have described studies of triclosan-induced bacterial strains or triclosan-tolerant bacterial isolates that have cross-resistance to multiple antibiotics. In the study by Curiao et al., triclosan exposure was used to create triclosan-tolerant strains of *E.coli* and *K. pneumoniae*. Subsequently, the differential expression of efflux pumps in the triclosan tolerant strains as compared to the susceptible strains was studied, and the genes *acrAB*, *acrF*, and *marA* were identified as being upregulated [13]. These genes have been associated with MDR strains of *Salmonella* [124]. In Aiello et al., 7 of 11 studies reviewed, demonstrated cross-resistance to at least one antibiotic, but the authors concluded that there was no correlation between the use of triclosan products and the presence of antibiotic-resistant bacteria among household members and their environment [125]. In a different review, clinical samples of methicillin-resistant *S. aureus* (MRSA) were effectively killed by triclosan; however, *Mycobacterium smegmatis* developed *inhA* mutations, which is also known to afford resistance to isoniazid. Additionally, *P. aeruginosa* and *E. coli* showed elevated resistance to several antibiotics [58,69].

Interestingly, other studies have shown that triclosan can potentiate the action of some antibiotics. Studies on *Rhodospirillum rubrum* show evidence that low levels of triclosan decreased the innate resistance to ampicillin and tetracycline but increased resistance to chloramphenicol and carbenicillin [126]. When an MDR *Acinetobacter baumannii* isolate was converted to a triclosan-tolerant strain, it exhibited increased sensitivity to minocycline, levofloxacin, and phosphonomycin (fosfomycin) [127].

Finally, additional research has been published in which *S. aureus* and *Enterococci* grown in subinhibitory concentrations, as well as *S. aureus* clinical isolates, showed no correlation between triclosan tolerance and the development of antibiotic resistance [57,73,115].

### 4.4. Metals

Cross-resistance has been described in the literature for strains exposed to low concentrations of metal salts, as well as in environmental and clinical isolates. In one example, cross-resistance to ciprofloxacin was described in *E. coli* that was exposed long-term to increasing concentrations of silver nitrate, although ciprofloxacin resistance was only identified in 1 out of 84 strains tested [74]. Additionally, cross-resistance has also been demonstrated in clinical and environmental isolates. Rojo-Bezares et al. observed metal tolerance in macrolide and/or lincosamide-resistant *Streptococcus agalactiae* strains isolated from pregnant women [128]. Timkova et al. identified antibiotic resistance in environmental isolates from a mine. These isolates showed metal tolerance to copper and antibiotic resistance to ampicillin and chloramphenicol [78]. Cross-resistance to ampicillin has also been described by Miloud et al., who isolated environmental species selected for ampicillin and observed resistance with other antibiotics in addition to silver and copper tolerance [77].

### 4.5. Chlorine-Releasing Agents

The literature shows a discrepancy in whether chlorine-releasing agents can induce antibiotic resistance. Lin et al. observed cross-resistance in *E. coli* exposed to a simulated low level of chlorination used in water treatment. The authors observed that cells were in a viable but non-culturable state, exhibiting reduced metabolic activity and enhanced viability when exposed to different antibiotics [129]. Cross-resistance to ciprofloxacin has also been observed in *K. pneumoniae* tolerant to sodium hypochlorite. Physiological changes were observed in the strains exposed to subinhibitory concentrations. On the other hand, other authors did not observe cross-resistance to antibiotics when laboratory-adapted *E. coli* and *Salmonella enteritidis* food isolates were exposed to sodium hypochlorite [104,130]. Similarly, Oggioni et al. investigated cross-resistance in 1600 clinical *S. aureus* isolates and observed no statistically significant correlation between susceptibility profiles for sodium hypochlorite and antibiotics [115].

### 4.6. Fixatives

Limited information was found on the evidence of cross-resistance of glutaraldehyde or formaldehyde-tolerant bacteria with antibiotics. Roedel et al. reported that in a panel of 93 *E. coli* isolates from broiler fattening farms, isolates with reduced formaldehyde susceptibility were rarely found, and that biocide tolerance was not interlinked with antibiotic resistance [111]. Piovesan et al. reported cross-resistance to chloramphenicol in an *E. coli* strain that showed reduced susceptibility to glutaraldehyde upon exposure to subinhibitory concentrations [104]. By contrast, other authors have reported that there is no evidence that glutaraldehyde can trigger cross-resistance with antibiotics [97,98].

### 4.7. Peroxygens

No cross-resistance to antibiotics has been described in the literature reviewed for peracetic acid [98,104]. Limited and conflicting information exists on the ability of hydrogen peroxide to induce cross-resistance with antibiotics. One study showed that some of the *E. coli* strains exposed to low concentrations of hydrogen peroxide in the laboratory exhibited changes in antibiotic susceptibility [104]. Wesgate et al. showed that long-term exposures to low concentrations of hydrogen peroxide were required to trigger an “unstable resistance” to ampicillin [131].

### 4.8. Alcohols

The literature shows limited to no evidence that alcohols lead to cross-resistance with antibiotics. Piovesan Pereira et al. did not observe cross-resistance to antibiotics when bacteria were exposed for approximately 500 generations to low concentrations of ethanol and isopropanol (4.25 and 2.5% *v*/*v*, respectively) [104]. Shan et al. studied the effectiveness of different antibiotics and disinfectants and concluded that alcohols had the fewest incidents of tolerance in clinically isolated strains of the seven biocides studied [132].

### 4.9. Iodine

The literature shows limited to no evidence that iodine can lead to cross-resistance with antibiotics [84,97,133,134]. Only one paper that we identified described some cross-resistance to medically relevant antibiotics in *E. coli* strains exposed to low concentrations of povidone-iodine [104].

## 5. Discussion

Several common themes emerged over the course of this review, which examined the state of the science on the impact of biocide use and the development of antibiotic resistance. First, the baseline logic driving the hypothesis seems to be that since subinhibitory levels of antibiotics can result in the emergence of antibiotic resistance and cross-resistance, it follows that subinhibitory levels of biocides may also induce antibiotic cross-resistance. Second, researchers have debated the relevance of these laboratory experiments to the real world in light of the variable persistence of biocides in the environment as well as the relative complexity of real-world environments. Finally, investigators have sought to understand the mechanisms behind laboratory-induced biocide tolerance and antibiotic cross-resistance. The main hypothesis for cross-resistance is focused on the function of efflux pumps, which are transport proteins involved in the export of toxic substances into their environment.

Substantial effort has been put toward investigating laboratory-induced biocide tolerance, followed by an assessment of antibiotic cross-resistance. In the reviewed studies for QACs, chlorhexidine, triclosan, and some metals, bacteria that acquired the ability to grow in the presence of increased biocide concentrations were identified after exposure to low concentrations. Most commonly, bacteria adapted to the biocide in the laboratory using sequential cultures of bacteria, starting at subinhibitory concentrations with increases in biocide concentration over time, after which cross-resistance to antibiotics was assessed. In these experiments, investigators often termed the bacteria as biocide-“resistant” whenever the biocide MICs increased. However, in many cases, the biocide tolerance level was still below the in-use concentration, leading to doubt that the biocide was “resistant” in real-world situations [135,136].

Moreover, the methods differed as to the generation of the biocide-tolerant bacteria (e.g., in a liquid culture or in a biofilm reactor) and any subsequent characterization. A lack of standardization of the experimental methods as well as the definition of “resistant” makes it challenging to assess the impact (if any) that low levels of biocide tolerance may have on the emergence or proliferation of antibiotic resistance. Without this standardization, much of the lab-based work remains difficult to link to a relevant clinical context [88]. Moreover, for bacteria strains that were adapted to increase biocide tolerance, the bacterial phenotype stability was rarely assessed. Knowing whether the adapted bacteria would be able to survive non-idealized laboratory conditions, as well as if the changes that confer biocide tolerance remain after the selection pressure is removed, are important questions that have not yet been addressed. Additionally, the stability of biocide tolerance varies, with said tolerance sometimes disappearing when the biocide pressure is removed, while at other times becoming permanent. In other cases, the biocide-tolerant bacteria may have a detrimental effect on fitness that would not allow them to compete with other bacteria to survive outside of a laboratory [13].

Translating the findings of model systems in the laboratory to real-world complexity is a common challenge in science. The laboratory studies on biocide tolerance and antibiotic cross-resistance have been conducted under idealized and controlled conditions, including culturing bacteria in growth media with defined concentrations of a single chemical stressor. In contrast, the real-world environment is significantly more complex, with bacteria growth in complicated matrices such as soil, food, wastewater, and in vivo, as well as the fact that some disinfectants use formulations that combine multiple biocide molecules with different mechanisms of action, making it more difficult for bacteria to develop tolerance [119]. To gain information about bacteria in their complex environments, researchers will typically study isolates and extrapolate their laboratory findings to what is understood about the real-world environment.

The stability of QACs, azoles, chlorhexidine, and metals in the environment has led to concerns that these biocide classes may persist in wastewater facilities from hospitals and food processing plants, as well as in run-off from agriculture. Several studies detected low levels of biocidal chemicals in wastewater, food, soil, mines, and other environmental sources [77,78,110,117,119]. Researchers hypothesize that biocide-tolerant bacteria rising from low concentration exposure in these niche environments may result in biocide-mediated antibiotic cross-resistance development in the real world. However, as discussed in Section 4, isolates with both biocide tolerance and antibiotic resistance have rarely been found. This may be in part due to the bioavailability of the biocides in these environments, which are likely quite different than in the laboratory experiments, due to biocides acting on and/or binding to other organic matter [137]. We did not identify any studies that considered this aspect. Finally, most bacterial isolates with identified “resistance” in the literature, as indicated by increased MIC values, remain susceptible to clinically used concentrations of disinfectants [135,136]. This finding is in contrast with antibiotic resistance, where the increasing bacterial MIC values are caused by concentrations much closer to the antibiotic dosages being used clinically, rendering certain antibiotics clinically obsolete [59].

In the investigation of potential mechanisms for cross-resistance, several hypotheses have gained traction. The most commonly proposed mechanism is the upregulation of gene expression for efflux pumps or increased efflux pump activity [69,96,98,103,104,110,117]. In studies where the efflux pumps were inhibited, the biocide-adapted, tolerant bacteria seemed to regain at least some susceptibility, but not in all cases [110]. Increases in efflux pump systems are used as an explanation for antibiotic cross-resistance in biocide-tolerant bacteria. Efflux pump regulation is one of the main systems that bacteria use to escape stressors in their environment, so it seems likely there are other factors involved with permanent adaptations to biocide tolerance.

When examining the possibility of efflux pumps conferring cross-resistance, it is important to connect three distinct elements. First, does exposure to a specific biocide result in the upregulation of an efflux pump gene? Second, are there examples where a specific bacterial species is shown to display this efflux pump mechanism of biocide resistance? Third, within the same species of bacteria, is a mechanism of resistance to antibiotics described using the same efflux pump? From our review, we identified efflux genes that met these three criteria. These genes included *AcrAB*, *CmeABC, EmrE*, *MdeA*, *MdfA*(*Cmr*/*CmlA*), *MepA*, *MexAB*, *MexCD*, *MexEF*, *NorA*, *NorB*, *QacABE*, and *QacE∆1* [97,138,139,140,141,142]. By satisfying the criteria, these genes could be at higher risk of conferring cross-resistance to antibiotics after biocide exposure. In these assessments, a fourth dimension should be evaluated: How do the measured changes in biocide tolerance and/or antibiotic resistance impact real-use settings? Can bacteria survive the recommended in-use concentrations of the biocides? Are those biocides used in the clinical context? Do the newly conferred antibiotic resistances require a change in treatment protocols clinically? These questions that are related to real-world relevance remain to be thoroughly explored in the literature, although some analyses along these lines suggest that there is minimal impact on the hospital environment [135,136].

## 6. Conclusions

While studies evaluating the linkage between biocide tolerance and antibiotic cross-resistance were identified, the evidence is not sufficient to establish a causal relationship between the two. Antibiotic cross-resistance was described for QACs, chlorhexidine, and metals, but the evidence was mostly based on laboratory experiments using subinhibitory concentrations significantly below the specified in-use concentrations. Just a few studies identified rare biocide-tolerant isolates that also showed antibiotic resistance. Conflicting evidence of antibiotic cross-resistance was found for chlorine-releasing agents, peroxygens, and triclosan. Limited to no evidence of antibiotic cross-resistance was found in the azoles, alcohols, fixatives, or iodine. No literature was identified that discussed antibiotic cross-resistance in relation to bronopol, ethylene oxide, or isothiazolinones. The primary mechanism proposed in the literature linking biocide tolerance and antibiotic cross-resistance is through efflux pumps. However, the link from laboratory studies to real-world contexts remains unclear, particularly with respect to any detrimental clinical impact. Moreover, it seems unlikely that a simple cause for such a linkage would exist since, in real-world situations, antibiotics often exist in complex environments, and the use of biocides is not expected to be the only or even the primary driving force for the occurrence of antibiotic resistance. Given the differing modes of action between biocides and antibiotics, which have highly specific biochemical activities in the target organism, it is anticipated that the broad-based physicochemical effects associated with most biocides would present a significantly higher evolutionary hurdle to the development of resistance. Both biocides and antibiotics are important tools in the arsenal of infection control against multi-drug-resistant bacteria; thus, the research community should continue to support studies that enable actionable data to inform policies that preserve these tools.

## Figures and Tables

**Figure 1 microorganisms-11-02000-f001:**
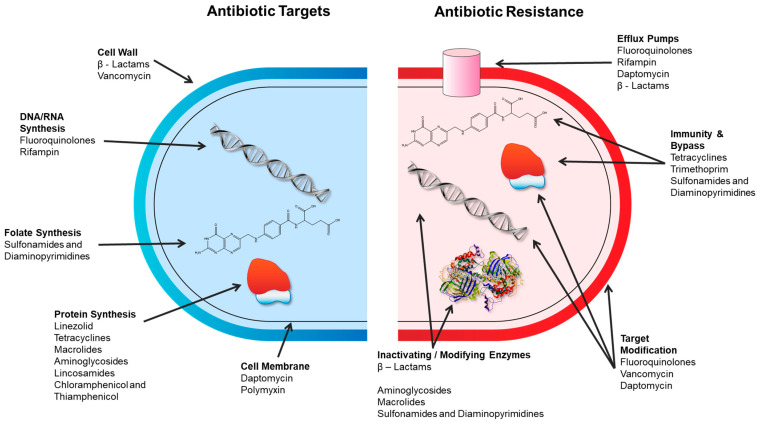
Antibiotic targets and mechanisms of antibiotic resistance. Adapted from [32].

**Table 1 microorganisms-11-02000-t001:** Chemistry and mode of action for various classes of antibiotics.

	Antibiotic Class	Representative Chemical Structure	Mode of Action
Drugs that Target Cell Wall Biosynthesis	β-Lactams: Penicillin Cephalosporins Carbapenems Monobactams	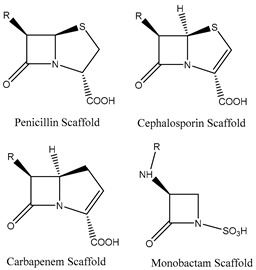	Inhibits the synthesis of the peptidoglycan layer of bacterial cell walls by binding to the active site of transpeptidases, known as penicillin-binding proteins (PBPs) [33]
	Glycopeptides and Lipoglycopeptides	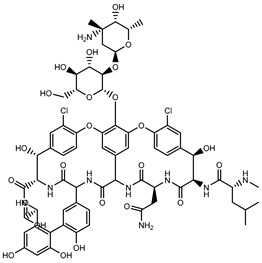 Vancomycin	Inhibits late stages of cell wall peptidoglycan synthesis by binding to precursors within the cell wall, preventing addition of new units to the peptidoglycan [29]
Drugs that Target Protein Synthesis	Aminoglycosides	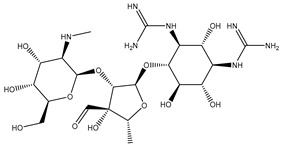 Streptomycin	Inhibits protein synthesis through high-affinity binding to the A-site of the 16S ribosomal RNA of the 30S ribosome [34]
	Tetracyclines and Alkylaminocyclines	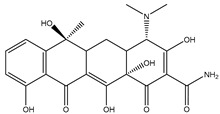 Tetracycline	Interferes with initiation step of protein synthesis by binding to the ribosomal 30S subunit thereby inhibiting binding of aminoacyl tRNA [29]
	Macrolides	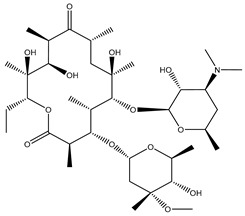 Erythromycin	Inhibits protein synthesis by binding to the peptidyl transferase center at the 50S surface, which causes multiple alterations of the 50S subunit functions [29]
	Lincosamides	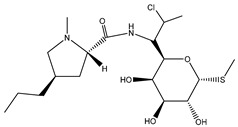 Clindamycin	Similar to macrolides [29,35]
	Chloramphenicol and Thiaphenicol	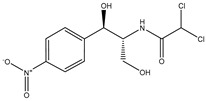 Chloramphenicol	Competitive inhibition for the binding of tRNA to the 50S peptidyltransferase domain. This triggers a conformational change in the ribosome that slows or inhibits aminoacyl tRNA incorporation [29]
	Oxazolidinones	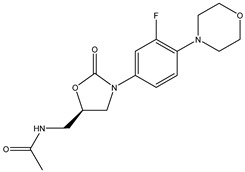 Linezolid	Inhibits protein synthesis by interfering with assembly of the initiation ternary complex of the 30S and 50S ribosomal subunits [29,36]
Drugs that Affect Nucleic Acids	Fluoroquinolones	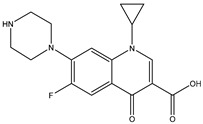 Ciprofloxacin	Inhibits the activity of topoisomerases [29]
	Ansamycins and Lipiarmycins	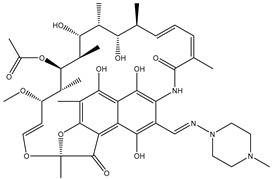 Rifampicin	Inhibits the initiation of DNA transcription by binding to the RNA polymerase or the DNA-RNA complex [29]
Antimetabolites	Sulfonamides and Diaminopyrimidines	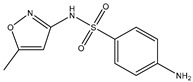 Sulfamethoxazole	Inhibits the folate pathway [29]
Drugs that Target the Membrane	Lipopeptides	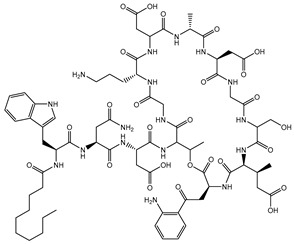 Daptomycin	Forms micelles (oligomeric assemblies) that interact with the membrane to cause a leakage of cytosolic contents [29,37]
	Cyclic Polypeptides (Polymyxins/Colistins)	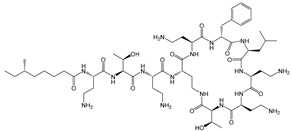 Polymyxin B	Acts as detergents and alters the permeability of the membrane [29,38]

**Table 2 microorganisms-11-02000-t002:** Chemistry and mode of action for various biocides.

	Biocide	Representative Chemical Structure(s)	Mode of Action
Ionic Interactions	Quaternary Ammonium Compounds (QACs)	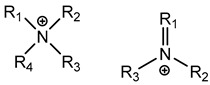 General QAC Structures 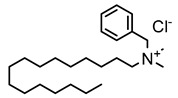 Benzalkonium chloride	Acts as a cationic detergent with electrostatic interactions with phospholipids [26,51,52,53,61,62]
Bisbiguanides	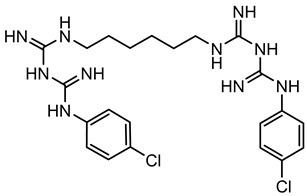 Chlorhexidine	Electrostatic interaction with phospholipids [26,52,53,62]
Hydrogen bond disruptors	Phenolics	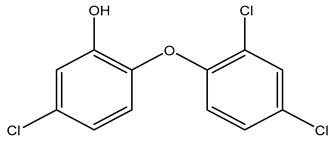 Triclosan	Not fully understood, but proposed to induce changes in membrane permeability and intracellular functions through hydrogen bonding [51,52,63] At low concentrations, triclosan acts as a site-specific inhibitor of enoyl-acyl carrier protein reductase [57]
Alcohols	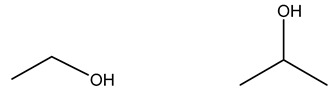	Solubilizes phospholipids and denatures proteins through disruption of hydrogen bonding [26,51,62]
Ethanol	Isopropanol
Chemical reactions	Metals	Ag	Interacts with thiol groups [26,53,54,62]
Chlorine-releasing agents	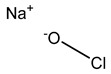 Sodium Hypochlorite	Halogenation of amino groups in proteins; oxidation of thiol groups [51,59]
Fixatives	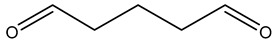 Glutaraldehyde 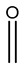 Formaldehyde	Alkylation of biomolecules with amino, imino, amide, carboxyl, and thiol groups (nucleophilic) [51,59]
Peroxygens	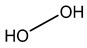	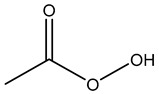	Oxidizing agents that produces hydroxyl free radicals that attack cell components, e.g., enzyme and protein thiols [26,51,53,54,62,64]
Hydrogen Peroxide	Peracetic Acid
Iodine	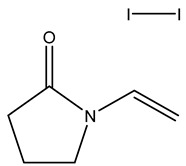 Povidone−−iodine	Oxidization of thiol groups on proteins, as well as oxidation of nucleotides and fatty acids [53,54,56,64,65]
Bronopol	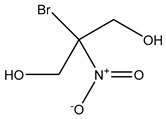	Oxidizes thiolcontaining materials and produces active oxygen species such as superoxide and peroxide [54,66]
Ethylene oxide	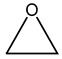	Alkylation of amino and thiol groups in proteins, as well as DNA and RNA [26,67]
Isothiazolinone	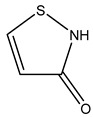	Acts as an electrophilic agent reacting with critical enzymes, reacting with thiols on proteins, and producing free radicals [26,55]

## Data Availability

Not Applicable.

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
