# Peer review of "Current Understanding of Potential Linkages between Biocide Tolerance and Antibiotic Cross-Resistance"

_microorganisms, 2023, doi:10.3390/microorganisms11082000_

Round 1
Reviewer 1 Report
This manuscript summarises current knowledge of the potential linkage between biocide tolerance and antibiotic cross-resistance. A description of the mechanisms of antibiotic action and resistance and also of biocide action and potential resistance mechanisms are also provided. This is an important topic, and it would be good to have an updated review on this topic. The current review is well-written; however, it lacks some detail in places. I have provided comments to improve the manuscript.
Line 30: Please replace the word ‘human contact sanitizers’ with ‘antiseptics’.
Line 38: Please rephrase the sentence starting with “Arguably of even greater impact,….”. This sentence is clunky and does not read well.
Line 47: Define MDR bacteria precisely i.e. when resistance is observed to at least one antibiotic from three different classes.
Line 49: “Bacteria have evolved a variety of strategies that have enabled resistance mechanisms…” Please rephrase to improve clarity.
Line 97 and throughout the manuscript: I am not particularly fond of the acronym AR. The term is not used very heavily so antibiotic resistance will be fine to use instead of AR.
Line 81: Gram-positive (with a capital) and Gram-negative.
This review would be improved by the addition of a figure to showing the different mechanisms of antibiotic resistance.
Table 1: Please check the mechanisms of action for tetracyclines and chloramphenicol as it is not clear from the description.
Table 1, middle column: Please correct rifampicin and sulfamethoxazole spacing to fit in one line.
Table 2, middle column: Please be consistent with the chemical drawings for things such as text size, line thickness etc.
Line 196 and onwards, section 3.1.1 Quaternary Ammonium Compounds (QACs). This section is not sufficient and needs to be rewritten. Avoid vague, meaningless statements such as “Quaternary ammonium compounds (QACs) are used for cleaning and deodorizing hard surfaces”. An excellent example of a QAC that is widely used is benzalkonium chloride that is used as a preservative in many pharmaceuticals such as nasal sprays, eye drops etc. It is also used in antiseptic in wet wipes.
Line 201: Please add reference 111 (Amsalu et al., 2020) here too as that study also showed the role of efflux pumps in benzalkonium tolerance.
Line 224, section 3.3.1 Metals: This section is too vague. The authors should consider discussing studies that looked into the transcriptomic responses to metals, e.g. https://doi.org/10.1016/j.scitotenv.2022.153915 and https://doi.org/10.3390/data7030035.
Line 243, section 3.3.3 Fixatives (aldehydes). This section needs to be updated. The authors are correct that formaldehydes are not used as biocides much more. However, there are a huge range of formaldehyde releasing agents such as bronopol, diazolidinyl urea and many more. These biocides are widely used in cosmetics and other pharmaceuticals and should be discussed here too. Bronopol is mentioned in a later section, but with no explanation of what class of chemical it is.
Line 252 – 253: Please provide a reference for this statement.
Line 259: What is SASP?
In the section about benzalkonium and resistance, please also discuss the results from doi: 10.1016/j.ebiom.2021.103653 on the effect of this QAC on resistance to the antibiotic gentamicin.
Lines 386 – 388: The authors completely misinterpreted this study. This study showed a link between benzalkonium resistance (this could be tolerance according to the definition from the current study) and ciprofloxacin resistance in P. aeruginosa isolated from healthcare wastewater specifically. This resistance was due to efflux pump activity as an efflux pump inhibitor reduced the MIC values for both benzalkonium chloride and ciprofloxacin to sensitive levels. Please correct this paragraph to accurately reflect the results from Reference 111.
Reference 28 is not referenced correctly. It seems to be a chapter in a book, but no book title, year of publication of publishing house is provided.
There is something wrong with references 61 and 62 as it is just an error message in the reference list.
The quality of English language is fine and only minor corrections are needed.
Reviewer 2 Report
1. In the abstract, it will be helpful to highlight the aim of the paper.
2. Please verify the punctuation in the manuscript, e,g., "...animal infections.[1,2]"; The punctuation mark "." is placed inappropriately.
3. Maybe you can summarize a little the conclusion section; it is quite extensive.
The paper is very good and exciting.
The manuscript requires minor English editing.
Reviewer 3 Report
The manuscript microorganisms-2454096 summarized the chemistries, modes of action, resistance/tolerance mechanisms for major classes of antibiotics and biocides, and discussed the potential links between biocide usage and antibiotic cross-resistance. The review was well written, with a well-organized abstract and introduction and a discussion. The main points and discussions of the review provided deep-understanding of linkages between biocide tolerance and antibiotic cross-resistance, and will be of extensive interests of antibiotic resistance research and control scientists. In my opinon, the manuscript may be acceptable in microorganisms.
